# Response of *Mycobacterium smegmatis* to the Cytochrome *bcc* Inhibitor Q203

**DOI:** 10.3390/ijms231810331

**Published:** 2022-09-07

**Authors:** Priyanka Chauhan, Santhe Amber van der Meulen, João Miguel Simões Caetano, Hojjat Ghasemi Goojani, Dennis Botman, Rob van Spanning, Holger Lill, Dirk Bald

**Affiliations:** Amsterdam Institute for Life and Environment (A-LIFE), AIMMS, Faculty of Science, Vrije Universiteit Amsterdam, De Boelelaan 1108, 1081 HZ Amsterdam, The Netherlands

**Keywords:** *Mycobacterium smegmatis*, terminal oxidases, Q203, electron transport chain

## Abstract

For the design of next-generation tuberculosis chemotherapy, insight into bacterial defence against drugs is required. Currently, targeting respiration has attracted strong attention for combatting drug-resistant mycobacteria. Q203 (telacebec), an inhibitor of the cytochrome *bcc* complex in the mycobacterial respiratory chain, is currently evaluated in phase-2 clinical trials. Q203 has bacteriostatic activity against *M. tuberculosis,* which can be converted to bactericidal activity by concurrently inhibiting an alternative branch of the mycobacterial respiratory chain, cytochrome *bd*. In contrast, non-tuberculous mycobacteria, such as *Mycobacterium smegmatis,* show only very little sensitivity to Q203. In this report, we investigated factors that *M. smegmatis* employs to adapt to Q203 in the presence or absence of a functional cytochrome *bd*, especially regarding its terminal oxidases. In the presence of a functional cytochrome *bd*, *M. smegmatis* responds to Q203 by increasing the expression of cytochrome *bcc* as well as of cytochrome *bd*, whereas a *M. smegmatis*
*bd*-KO strain adapted to Q203 by increasing the expression of cytochrome *bcc*. Interestingly, single-cell studies revealed cell-to-cell variability in drug adaptation. We also investigated the role of a putative second cytochrome *bd* isoform postulated for *M. smegmatis*. Although this putative isoform showed differential expression in response to Q203 in the *M. smegmatis*
*bd*-KO strain, it did not display functional features similar to the characterised cytochrome *bd* variant.

## 1. Introduction

Tuberculosis continues to present a serious public health challenge with 1.5 million deaths in 2020 [1]. The WHO ‘End TB’ strategy aims at momentous reduction in both TB deaths and TB incident rates by 2030 [1]. However, TB treatment remains challenging due to a variety of reasons, including genotypic and phenotypic drug resistance, bacterial defence factors and adaptation of bacterial metabolism, which pose serious threats to the success of existing TB drug regimens. Therefore, the design of effective combination regimen remains imperative for improving TB chemotherapy.

The regulatory approval of the ATP synthase inhibitor bedaquiline (BDQ) [2,3,4], the first within four decades for a TB drug, has marked energy metabolism as a compelling target in *Mycobacterium tuberculosis* for the development of next-generation anti-TB drugs [5,6,7,8]. Next to small molecules targeting ATP synthase, inhibitors interfering with components of the mycobacterial respiratory chain have been identified and are currently evaluated in clinical trials.

The mycobacterial respiratory chain is branched; electrons from the menaquinol pool can be transferred either to the cytochrome *bcc*-*aa*_3_ branch or to the alternative cytochrome *bd* branch [5,6,7,8]. Both enzymes act as terminal oxidase, reducing molecular oxygen to water. Comparison to homologous terminal oxidases in *Escherichia coli* suggests that cytochrome *bcc*-*aa*_3_ represents an energy-efficient terminal oxidase with low affinity for oxygen, whereas cytochrome *bd* is less energy-efficient but shows higher oxygen affinity and lower sensitivity to inhibitors such as nitric oxide and cyanide [9,10,11,12]. Targeting these terminal oxidases currently is regarded as a viable therapeutic strategy and has attracted strong attention in the TB drug discovery field [10,11,12,13,14].

Whereas small-molecule inhibitors of mycobacterial cytochrome *bd* have only recently been described [15,16,17,18], multiple cytochrome *bcc* inhibitors have been discovered. Among them, Q203 is active on drug-sensitive, multi-drug-resistant, and extremely drug-resistant strains of *M. tuberculosis* [19], and presently is evaluated in phase-2 clinical trials [20]. The antimicrobial spectrum of Q203 revealed excellent potency against *M. tuberculosis* H37Rv (MIC_90_ ~10 nM), but no apparent activity against non-tuberculous mycobacterial strains such as *M. smegmatis* [19].

For the design of an efficient next-generation tuberculosis chemotherapy regimen, insight into bacterial defence against (candidate) drugs is required. In this regard, various studies have shed light on the physiological response of *M. tuberculosis* to an impaired cytochrome *bcc*-*aa*_3_ respiratory branch. Genetic inactivation of cytochrome *bcc* or chemical inhibition by small molecules results in pronounced upregulation of the alternative cytochrome *bd* branch [21,22]. As a result, Q203 acts bacteriostatic against *M. tuberculosis* [15,16,23,24] and bactericidal activity is only achieved upon simultaneous inactivation of cytochrome *bd* [15,24]. Less information is available regarding the response to cytochrome *bcc* inhibitors in *M. smegmatis*. In this bacterium, genetic inactivation of cytochrome *bcc* resulted in diminished growth and upregulation of cytochrome *bd* [25].

In the current report, we investigate the response of *M. smegmatis* to Q203, both in the absence and in the presence of a functional cytochrome *bd* pathway.

## 2. Results

### 2.1. Mycobacterium smegmatis bd-KO Can Outgrow Q203

Whereas growth of *M. tuberculosis* is strongly suppressed by Q203 (MIC~10 nM) [19], *M. smegmatis* is insensitive to Q203, with a minimum inhibitory concentration (MIC) of >50 µM [15,19,26]. We confirmed this lack of sensitivity (Appendix A) and found that Q203 did not affect the length of the lag phase, the doubling time or the final growth yield of *M. smegmatis* wild-type (WT) (Figure 1A,B). As indicated by several studies, the susceptibility of various mycobacterial strains for Q203 can be increased by genetic or drug-based inactivation of cytochrome *bd*, an alternative terminal oxidase that can compensate for inactivation of the cytochrome *bcc* complex [15,16,24,26,27]. In case of *M. smegmatis*, in the absence of functional cytochrome *bd* the MIC_90_ for Q203 decreases to 2.5 μM ([15,26], see also Appendix A).

Interestingly, we observed that a *M. smegmatis bd*-KO strain [28] eventually can overcome growth inhibition by Q203, even at a concentration of 4X MIC (10 µM) (Figure 1C). At this concentration, Q203 considerably extended the lag-phase, but subsequently the bacteria commenced growth with doubling time and final growth yield unaltered as compared to the bacteria in the untreated control (Figure 1C,D). Bacterial growth parameters remained unchanged when Q203 was incubated in the assay medium for 24 h prior to treating the bacteria, indicating that the outgrowth was not the result of drug degradation (Appendix A).

### 2.2. Outgrowth of M. smegmatis bd-KO in Presence of Q203 Is Not Due to Selection of Resistant Mutants

The outgrowth observed above for the *M. smegmatis bd*-KO strain may be due to acquisition of drug resistance during treatment, as has been observed for the front-line drug isoniazid [29]. Therefore, we collected *M. smegmatis bd*-KO samples that outgrew the effect of Q203 and re-treated the bacteria again with the drug. Upon re-treatment the duration of the lag phase decreased from ~25 h to ~18 h before exhibiting the outgrowth (Figure 2), suggesting that the bacteria acquired some adaptive advantage from the first treatment. However, re-growing the bacteria in drug-free medium for 16 h prior to re-treatment led to a complete loss of this adaptive advantage and the bacteria exhibited an even longer lag before the outgrowth (Figure 2). This observation indicates that the “adaptive advantage” does not represent genotypic resistance but is the result of a phenotypic and semi-heritable feature.

### 2.3. Phenotypic Cell-to-Cell Variability in Adaptation of M. smegmatis bd-KO to Q203

The pronounced lag-phase after the exposure of *M. smegmatis bd*-KO cells to Q203 may result from a stable balance between growing and dying cells. To test this, we counted the colony-forming units (CFUs) for *M. smegmatis* in the presence of Q203. As, expected, Q203 had no influence on the viability of *M. smegmatis* WT during the observed 48 h time frame (Appendix A). In the case of *M. smegmatis bd*-KO, the bacterial counts showed only a minimal change during the initial 24 h period of treatment with Q203, but strongly increased afterwards (Figure 3A,B), consistent with the results from the growth curve analysis above. The results, therefore, gave no indication of cell death or for the presence of subpopulations with differential sensitivity for Q203 being present at the beginning of the treatment period.

Adaptation to a drug can either be a population-wide phenomenon or individual cells can exhibit differences in the adaptability. The traditional microbiological assays, which are principally population-averaging methods, often mask the heterogeneous behaviour of the cells. Therefore, to get a deeper insight into the adaptability of *M. smegmatis* to Q203, we utilised agarose-pad-based time-lapse microscopy to observe individual bacterial cells. In the case of *M. smegmatis* WT treated with Q203, no significant difference in the mean growth rate was observed as compared to untreated control (Figure 3C). Interestingly, *M. smegmatis bd*-KO treated with Q203 showed considerably increased cell-to-cell variability while adapting to Q203, both compared to the WT strain in presence of Q203 and compared to the *bd*-KO strain in absence of Q203 (Figure 3C). This result suggests that the observed lag phase in the case of *M. smegmatis bd*-KO cells treated with Q203 is highly dynamic.

### 2.4. Expression Changes of Respiratory Branch Genes in the Presence of Q203

We utilised RT-qPCR analysis to gain insight into how the expression profile of genes encoding the terminal oxidases is remodelled upon treatment with Q203. As described previously, genetic impairment of the cytochrome *bcc* complex in *M. smegmatis* leads to the adaptation of the respiratory network by re-routing the electron flux through cytochrome *bd* [25]. A similar respiratory flexibility is established in *M. tuberculosis*, where drug-based inhibition of cytochrome *bcc* induced cytochrome *bd* [22], facilitating bacterial survival. Extending these previous experiments, we assessed the expression levels of the *cydA* gene, encoding the CydA subunit of cytochrome *bd*, and of the *qcrB* gene, encoding the QcrB subunit of cytochrome *bcc*, as representative genes. As the results from our growth inhibition experiments suggest that the sensitivity for Q203 may depend on the bacterial growth state, we quantified expression of these genes at various phases of growth both in the presence and absence of Q203.

Upon treatment of *M. smegmatis* WT with Q203, RNA transcripts of *cydA* were strongly induced as compared to untreated samples (Figure 4A). The upregulation of *cydA* was found to be significant (*p* value ≤ 0.01) across the growth phases investigated, with exception of early stationary phase (Figure 4A). We also evaluated if *M. smegmatis* WT additionally employs the strategy of upregulating the drug’s target, thereby out-titrating the effect of Q203 or, alternatively, if this respiratory branch is downregulated and compensated for by induced cytochrome *bd*. Indeed, we found that treatment with Q203 triggered induction of the *qcrB* gene (Figure 4B). Upregulation of *qcrB* was significant across the growth phases investigated (*p* value ≤ 0.01), with exception of the stationary phase (Figure 4B).

Next, we evaluated how the bacteria respond to Q203 if the alternative cytochrome *bd* pathway is not available. We found that in the *M. smegmatis bd*-KO strain in the presence of the drug the *qcrB* transcript levels are increased as compared to the untreated control (Figure 4C). However, the degree of induction was indistinguishable from that of the WT strain (Figure 4C). The absence of the alternative cytochrome *bd* respiratory branch, combined with the observed inability of the *bd*-KO strain to further increase the levels of cytochrome *bcc* may largely explain the higher sensitivity for Q203 as compared to *M. smegmatis* WT.

### 2.5. The Putative Second Cytochrome bd in M. smegmatis

Bacterial respiratory chains can contain multiple terminal oxidases, in particular, more than one cytochrome *bd*-type terminal oxidase can be present [30,31,32,33,34,35,36]. *M. tuberculosis* has only one *bd*-type terminal oxidase, which is encoded by the *cydAB* operon [28,37]. For *M. smegmatis*, cytochrome *bd* encoded by *cydAB* operon and with the subunit composition CydAB has been microbiologically, biochemically, and structurally characterised ([15,38], Figure 1, Figure 2, Figure 3 and Figure 4). In addition, based on comparative genome sequence analyses, the existence of a second cytochrome *bd*-type terminal oxidase in *M. smegmatis* has been suggested [28]. The genes encoding this putative second cytochrome *bd* variant were annotated as MSMEG_5584 and MSMEG_5585 and were reported as homologues of the *Bacillus subtilis ythAB* genes [28]. We found that MSMEG_5584 and MSMEG_5585 in the Mycobrowser database are described as a short-chain dehydrogenase and a hypothetical protein with unknown function, respectively. Pairwise sequence alignment of MSMEG_5584 with *M. smegmatis CydA* and of MSMEG_5585 with *M. smegmatis CydB* showed <10% sequence identity on the protein level (Appendix A). Likely due to a re-annotation of the *M. smegmatis* genome, the genes now referred to as MSMEG_5584 and MSMEG_5585 are unlikely to play a role as terminal oxidase.

Using the NCBI Genome database, we found that next to the *cydAB* operon, a second operon encoding a putative cytochrome *bd*. These genes, referred to as MSMEG_5605 or *appC* and MSMEG_5606 or *appB*, are predicted to encode the two major subunits AppC (subunit I) and AppB (subunit II) of this putative second cytochrome *bd* complex (Figure 5A). We confirmed the presence of these annotated genes in the genome (Appendix A) and the presence of the gene transcripts (Figure 5B).

The observed amino acid sequence identity was 26.2% between CydA and AppC and 22.1% between CydB and AppB (Appendix A). Multiple sequence alignment of *M. smegmatis* CydA, *M. smegmatis* AppC, and their counterparts from *M. tuberculosis* and *E. coli* showed conservation of several amino acid residues previously described as representative and critical for cytochrome *bd* function (Figure 5C). Cytochrome *bd*-type terminal oxidases oxidise a quinol substrate and transfer the electrons via enzyme-bound heme groups onto molecular oxygen, a reaction coupled to the uptake of protons from the cytosol [9]. Conserved residues include AppC His33, Glu113, His197, and Met417 (Figure 5C, marked with red boxes), which in *E. coli* CydA (His19, Glu99, His186, and Met393) are reported as heme-binding residues [39,40,41,42], AppC Ala111, Leu115, and Phe118 (marked with a green box), which in *E. coli* CydA (Ala97, Leu101, and Phe104) are proposed as part of a putative oxygen access channel [41,42] and AppC Glu121 (marked with an orange box), which *in E. coli* CydA (Glu107) is thought to contribute to proton access from the cytosol to heme d [41,42,43]. In addition, *M. smegmatis* AppC has the highly conserved residues AppC Lys275 and Glu280 (blue box), which in *E. coli* CydA (Lys252, and Glu257) are located in a periplasmic loop involved in binding of the quinol substrate (Q-loop) and which are essential for quinol oxidation [44,45,46] (Figure 5C). Similar to CydA from *M. smegmatis* and *M. tuberculosis*, *M. smegmatis* AppC does not harbour the insert of ~60 residues found in CydA and AppC from *E. coli* and other proteobacteria that distinguishes two cytochrome *bd* families [9,47,48] and is important for the stability of *E. coli* CydA [49,50] (Appendix A). Next to AppC, *M. smegmatis* AppB features residues that are conserved between mycobacterial and *E. coli* cytochrome *bd* variants (Appendix A). Thus, AppC and AppB share characteristic features of canonical cytochrome *bd* subunits, providing additional support for the previous notion that *M. smegmatis* harbours a second *bd*-type terminal oxidase [28]. We will use the term “putative AppCB variant” to describe this putative, second *M. smegmatis* cytochrome *bd* variant, as opposed to the validated variant CydAB.

### 2.6. Differential Expression of the Putative Second Cytochrome bd in M. smegmatis

The function of this putative second cytochrome *bd* variant in *M. smegmatis* is unknown. As it displays canonical features of *bd*-type terminal oxidase, we reasoned that expression of *appCB* may be responsive to inactivation of cytochrome *bcc*. In *E. coli*, the two cytochrome *bd* variants present each displayed clearly distinct expression patterns in response to environmental stressors [34]. We therefore assessed if the expression of the putative AppCB variant in *M. smegmatis* mirrors the expression of the validated CydAB variant. In contrast to our results for expression of the *cydA* gene (Figure 4A), the *appC* gene in *M. smegmatis* WT was not induced by Q203, with the exception of a moderate upregulation observed in the mid-exponential growth phase (Figure 6). Upon the inactivation of cytochrome *bcc*, the bacteria employ the strategy of upregulating the CydAB variant and not the putative AppCB variant. In contrast, the *bd*-KO strain displayed pronounced upregulation of *appC* upon treatment with Q203 across all growth phases, in particular at the stationary phase of growth (Figure 6). Upregulation of *appC* was found to be significant when compared to untreated control for all growth phases investigated (*p* < 0.001). Induction of *appC* in the *bd*-KO strain was also found to be significantly higher as compared to *appC* induction in the WT strain, with exception of early stationary phase (Figure 6). Thus, under conditions when the function of cytochrome *bcc* is challenged and the CydAB alternative branch is not available, the putative third terminal oxidase, AppCB, is induced.

### 2.7. The Putative Terminal Oxidase AppCB Does Not Have the Same Properties as CydAB

We next assessed oxygen consumption by live *M. smegmatis* bacteria using an Extracellular Flux Analyser. This approach has previously been used to monitor the respiratory activity of *M. tuberculosis* and its response to drugs interfering with respiration [23]. We initially cultured *M. smegmatis* in the absence of Q203, immobilised the bacteria onto the well-plate of the Extracellular Flux Analyser, and monitored the oxygen consumption rate (OCR). In the case of *M. smegmatis* WT, we observed a continuous increase of the OCR after the addition of glucose as energy source (Figure 7A). Addition of Q203 to the immobilised bacteria triggered a further increase of the OCR (Figure 7A). A similar respiratory activation upon Q203 treatment has been reported earlier for *M. tuberculosis* and has been attributed to re-routing of electron flow via the cytochrome *bd* (CydAB) branch [23]. The activation observed here for *M. smegmatis* appeared less pronounced than the activation previously measured for *M. tuberculosis*, mainly due to the continuous activity increase already observed for *M. smegmatis* WT in the absence of Q203 (Figure 7A). The OCR of *M. smegmatis bd*-KO in the absence of Q203 resembled the activity of the WT strain, however, upon addition of Q203, the OCR of the *bd*-KO strain considerably decreased (Figure 7A). Interestingly, in the case of *M. smegmatis*
*bd*-KO, the OCR was not completely abolished by Q203 (Figure 7A,) in contrast to data reported for *M. tuberculosis* [23]. We also evaluated how the respiratory activity of *M. smegmatis* responds to Q203 when the bacteria were grown in the presence of this drug. Under these conditions, no respiratory change upon addition of Q203 was visible for *M. smegmatis* WT (Figure 7B), but the *bd*-KO strain displayed a decreased OCR, with a certain degree of residual activity (Figure 7B). Hence, in this assay, the AppCB variant clearly did not facilitate the respiratory activation that was observed for cytochrome *bd* in *M. tuberculosis* and to a lower degree for the validated cytochrome *bd* variant CydAB in *M. smegmatis*. On the other hand, these results do not exclude that AppCB might be involved in maintaining a low, basic OCR.

We also evaluated if the presence of the putative second cytochrome *bd* variant can be detected based on characteristic heme absorption peaks in the UV-VIS absorption spectrum [9]. We isolated membrane fractions from *M. smegmatis* grown to stationary phase, the growth state with the highest detected expression level of the *appC* gene in the *bd*-KO strain (see Figure 6). The heme spectra of membrane fractions from *M. smegmatis* WT revealed a clearly distinguishable absorption peak at ~630 nm typical for heme d in cytochrome *bd* (Figure 7C), as described earlier [28]. This characteristic peak further increased when the bacteria were grown in the presence of Q203 prior to the isolation of the membrane fractions (Figure 7D), consistent with our transcriptional data and with previous transcriptional studies upon genetic inactivation of cytochrome *bcc* [25]. As expected, the heme *d* peak was absent in the untreated *M. smegmatis bd*-KO strain (Figure 7E, [27]). After culturing the *bd*-KO strain in the presence of Q203, no clear peak was discernible between 620–660 nm, although a minor shoulder was observed in that wavelength range (Figure 7F).

Taken together, under the conditions tested here, the putative second cytochrome *bd* variant AppCB exhibited markedly different properties as compared to CydAB.

## 3. Discussion

In order to fully exploit the potential of an antibacterial drug and to design efficient drug combinations, strategies that bacteria might employ to escape the drug’s impact need to be elucidated. Q203 and other small-molecule inhibitors targeting cytochrome *bcc* have been extensively characterised against the highly sensitive pathogenic *M. tuberculosis*, but data on the less sensitive model strain *M. smegmatis* strain are scarce.

Upon inactivation of cytochrome *bd* in *M. smegmatis*, the sensitivity to Q203 increases, but the drug is limited to a bacteriostatic effect, as represented by our CFU analysis. In contrast, cytochrome *bcc* inhibitors act strongly bactericidal against a cytochrome *bd* mutant of *M. tuberculosis* H37Rv [22]. Similarly, Q203 in combination with a cytochrome *bd* inhibitor is bactericidal against wild-type M. tuberculosis [15,16]. Interestingly, analysis of growth of the *M. smegmatis bd*-KO strain over time in presence of Q203 revealed an extended lag phase, followed by growth resumption with a doubling time comparable to untreated bacteria, apparently without selecting for drug-resistant mutants. Similar outgrowth in presence of cytochrome *bcc* inhibitors has previously been reported for M. tuberculosis; however, the outgrowth was restricted to laboratory-adapted strains with functional cytochrome *bd* and was not detected in a *M. tuberculosis bd*-KO strain [22]. Apparently, the MIC_90_ value determined for the *M. smegmatis bd*-KO strain by an established microbiological assay using resazurin reduction does not fully describe growth inhibition by Q203. For the *M. smegmatis bd*-KO strain, analysis of the outgrowth in presence of Q203 at single-cell level by time-lapse microscopy revealed some cells adapting faster than others. Phenotypic heterogeneity by variability in expression and activity of important functions of bacterial cells may well contribute to the adaptive response in presence of a drug [51].

An extended lag phase in the presence of a drug might be needed to devise an appropriate respiratory/metabolic response. Bacteria can respond to an antibacterial drug in various manners. The upregulation of a drug’s target represents a strategy to out-titrate the inhibitory effect of the drug. As an example, the response of *M. tuberculosis* to the ATP synthase inhibitor bedaquiline involves pronounced induction of ATP synthase, which may in part play a role in delaying the bactericidal effect of this drug [52]. Our results reveal a similar response for *M. smegmatis* in presence of Q203, although the degree of induction was moderate. A comparable moderate induction of the *qcr*A and *qcr*C genes was previously reported for *M. tuberculosis* in presence of Q203 [16]. Q203 also triggers pronounced upregulation of the *cydA* gene in *M. smegmatis*, mirroring the response to genetic or drug-based inactivation of cytochrome *bcc* reported earlier [22,25]. Hence, our results suggest that *M. smegmatis* employs a double-pronged strategy to overcome the effect of Q203, including both the increased production of the target enzyme cytochrome *bcc* and the induction of the alternative cytochrome *bd* respiratory branch.

The presence of a second *bd*-type terminal oxidase in *M. smegmatis* was proposed based on genomic analysis [28]. Usage of multiple cytochrome *bd* variants is not restricted to *M. smegmatis* and may assist bacteria with the adaptation to fluctuating environmental conditions. In *E. coli*, where two cytochrome *bd* variants are characterised, the AppCB variant appears important for the transition from anaerobiosis to aerobiosis [36,53] and for survival in the mammalian host during gut inflammation [35]. In *Salmonella enterica* serovar Typhimurium, a critical role for growth under the oxygen-limited conditions encountered in the antibiotic-treated gut has been proposed for the second cytochrome *bd* variant [33,54]. Our results reveal that the expression patterns of the validated cytochrome *bd* variant (CydAB) and the putative second variant (AppCB) differ in their response to the cytochrome *bcc* inhibitor Q203. An earlier study reported that MSMEG_5584 and MSMEG_5585, unlike *cydA*, were not induced in response to genetic inactivation of cytochrome *bcc* [25], in line with our data on drug-based inactivation. Here, our results indicate that *appC* can indeed be induced if cytochrome *bcc* is inactivated and no functional CydAB is present. On the other, hand, our examination of AppCB heme fingerprints and oxygen consumption activity did not provide compelling evidence for a function of AppCB as cytochrome *bd*-type terminal oxidase. Here, further experimentation at biochemical level is needed to clarify the enzymatic function and physiological importance of AppCB in *M. smegmatis*.

The expression of terminal oxidase genes in *M. smegmatis* in response to the bacterial growth phase has been described earlier [55]. In our current study, we extend this previous work and examine the growth-phase-dependent transcriptional response to an antibacterial drug. The growth-phase-dependent induction determined for the drug’s target cytochrome *bcc* and for the alternative cytochrome *bd* pathways suggests that selection of the growth state might be carefully considered when characterising drug candidates. Clearly, further experimentation is required to characterise the mycobacterial terminal oxidases and to evaluate the impact of their expression on drug efficacy. Furthermore, *M. smegmatis* may employ additional strategies to circumvent the effect of Q203 that are not covered by our study and need to be explored. This may include higher expression and/or more efficient use of efflux pumps that can decrease the intracellular drug concentration or efficient usage of the glycolytic pathway that may compensate for missing respiratory chain function.

## 4. Materials and Methods

### 4.1. Bacterial Strains, Culture Conditions and Reagents

A *Mycobacterium smegmatis* mc2 155 strain lacking cytochrome *bd* (*cydA*::aph, referred to as “*bd*-KO” in this report) [28] was provided by Prof. Bavesh Kana (University of Witwatersrand) and Prof. Valerie Mizrahi (University of Cape Town). *Mycobacterium smegmatis* mc2 155 (*M. smegmatis* WT) and the *bd*-KO strain were cultured at 37 °C at under shaking conditions in Middlebrook 7H9 medium containing 0.05% Tween-80 supplemented with 10% Middlebrook ADC Enrichment (7H9 complete media). When needed, Kanamycin (Kan, 50 µg/mL) was used to select for mutant strains.

### 4.2. Growth Curve Analysis

For the growth analysis, mid-logarithmic phase (OD_600_~0.4) culture of *M. smegmatis* WT and *bd*-KO were diluted to OD_600_~0.0015 in 7H9-ADS-0.05% Tween-80 media and treated with three different concentrations (100 nM, 1 μM and 10 μM) of Q203 (MedChemExpress (Monmouth Junction, NJ, USA)) in 96-well flat-bottom clear bottom microplate (CellStar™, Greiner Bio-One™ (Alphen aan den Rijn, The Netherlands)). Suitable controls included wells containing solvent DMSO control/bacteria only/ media only. The plates were incubated under shaking condition at 37 °C in a spectrophotometer (SpectraMax Plus 384, Molecular Devices (San Jose, CA, USA)) for 96 h and the OD_600_ was measured at various time intervals. Growth of *M. smegmatis* WT and *bd*-KO was additionally measured in 50 mL flasks in the presence and absence of 10 μM Q203 (Appendix A). Growth phases were determined based on curve analysis using exponential (nonlinear) regression function. The OD values are first converted to ln OD and then plotted against time. The time points window whose OD_600_ values allowed for a fit with a R^2^ value > 0.99 were regarded as exponential phase, time points before that interval as lag-phase and after that interval as stationary phase. Within the exponential phase, the first time point of the fit was regarded as early exponential phase, and the time point with average OD_600_ was regarded as mid-exponential phase. Within the stationary phase, the first time point that deviated from the exponential regression fit (or result in non-linearity) was regarded as early stationary phase and the time point half-way between start of the stationary phase and end of the measurement was regarded as late-stationary phase [56].

### 4.3. Determination of Minimal Inhibitory Concentration (MIC)

The resazurin microtiter assay (REMA) was performed as previously described [57]. *M. smegmatis* WT and *M. smegmatis* bd-KO were cultured in Middlebrook 7H9 broth, containing 0.05% Tween-80 supplemented with 10% BD Middlebrook ADC Enrichment and two-fold serial dilution of Q203 in 96-well plate (initial OD_600_ 0.0015). Following the 48 h incubation period at 37 °C, wells were supplemented with 30 µL 0.02% filter-sterilised resazurin and 12.5 µL 20% filter-sterilised Tween-80 (alamar blue solution) and incubated overnight. The MIC was determined as the lowest Q203 concentration that prevented this colour change (visually observed)

### 4.4. Time-Lapse Microscopy

The agarose pads were prepared using low-melting point agarose (1.5% (*w*/*v*)) dissolved in 7H9-0.05% Tween-80 media. After agarose was cooled down to around 40–50 °C, 10% of ADC enrichment was added. Gently, 700 µL of agarose was dispensed onto the glass cover slip (18 × 18 mm) placed in sterile petri dish on the flat surface. The second cover clip was placed on the top to create an agarose sandwich. The petri dish was closed and left for ~30 min to let the agarose solidify. The upper coverslip was then removed gently using a sterile scalpel blade and the required size of agarose pad was cut out. Then, the 2 µL of bacterial sample (~4000 cells/µL) was pipetted on top of the pad. After allowing cells to be adsorbed for 5–10 min, the agarose pad was flipped onto the glass slide. To image multiple samples at the same time, the 8-well glass slide was used and several agarose pads were flipped onto the slide. The lid of the slide was closed and sealed with parafilm to minimise drying of agarose pads. A drop of immersion oil was placed on the 100X objective and the slide was mounted on the microscope stage. The imaging was carried out in time-lapse manner. When needed, kanamycin and Q203 were added into the media while preparing the agarose pads. For untreated control, DMSO was added in the same percentage of as of Q203-treated cells. Please note that the number of cells used for MIC calculation differ from what is applied on agarose pad; hence, the concentration of Q203 for the agarose pad experiment was adjusted accordingly.

Colonies that were sufficiently separated from other colonies were selected by hand, after which the colonies were segmented using ImageJ [58] with the Weka segmentation plugin [59] and colony sizes were determined for each frame. Next, growth rates in time for each colony were determined by fitting a linear regression on the ln-transformed colony size using a sliding window of 4 frames. Finally, growth rates (i.e., the median value of the obtained slopes over time) were determined for each colony.

### 4.5. Determination of Colony Forming Units (CFU)

To assess the killing of bacteria by Q203, cultures (mid-exponential phase, ~0.4 OD) were diluted to OD = 0.0015. The tested concentrations of Q203 were then added to the culture and incubated at 37 °C under shaking conditions. At the indicated antibiotic exposure time, the samples were collected and serially diluted (10-fold, 10^0^–10^−6^) and spotted on 7H11 agar plates supplemented with 10% (vol/vol) ADC enrichment (Albumin-Dextrose-Catalase, Difco, Forn El Chebbak, Lebanon) and 0.4% activated charcoal. The plates were incubated for 3–4 days at 37 °C to determine colony-forming units (CFU) counts.

### 4.6. RNA Isolation and Reverse Transcriptase qPCR Analysis

RNA isolation was performed using the NucleoSpin RNA Mini kit (Machery Nagel (Düren, Germany)). Briefly, 10^9^ cells were harvested from *M. smegmatis* WT and *bd*-KO cultures at various growth phases (Q203 treated or untreated) and RNA was isolated according to the manufacturer’s instructions, with some modifications. To lyse the cells, BeadBug™ tubes prefilled with 0.1 mm Zirconium–silica glass beads were used. The tubes containing cells were bead-beaten with 500 μL Buffer RA1 and 5 μL β-mercaptoethanol for 1 min at a speed of 6 m/s. RNA concentration was determined using the Qubit™ RNA HS Assay Kit (Invitrogen (Waltham, MA, USA)), as described in the manufacturer’s manual. 200 ng of DNA free RNA was reverse transcribed to cDNA using High-Capacity cDNA Reverse Transcription Kit with RNase Inhibitor (Applied Biosystems^®^ (Waltham, MA, USA)) according to the manufacturer’s protocol with one adaption. Denaturation of the sample, with all the reaction components except reverse transcriptase (RT), was done at 70 °C for 10 min and chilled immediately on ice for 5 min. The mix was then incubated at 25 °C for 10 min followed by addition of 50U of RT. The obtained cDNA was subjected to qPCR analysis using gene-specific primers (Appendix A), SensiFAST™ SYBR^®^ Hi-ROX Kit and 10% DMSO (in Applied Biosystems 7300 real-time PCR system, Thermo Scientific (Waltham, MA, USA)) using the following temperature conditions: 94 °C (2 min) followed by 40 cycles of 94 °C (20 s), 57.0 °C (30 s) and 72 °C (30 s). The transcript levels of individual genes in various RNA samples were normalised using 16S rRNA.

### 4.7. Identification of Genes Encoding a Putative Second Cytochrome bd Variant

Identification of *appCB* genes encoding a second putative cytochrome *bd* in *Mycobacterium smegmatis* mc2 155 was performed with the online tool NCBI genome ((https://www.ncbi.nlm.nih.gov/gene/45746913, accessed on 22 November 2021) and (https://www.ncbi.nlm.nih.gov/gene/66736902, accessed on 22 November 2021)). Pairwise sequence alignment was performed using the online tool EMBOSS Needle72 (https://www.bioinformatics.nl/cgi-bin/emboss/help/needle, accessed on 22 November 2021). Multiple sequence alignment was performed in the program Jalview73, using sequences downloaded from the UniProt74 database. Subsequently, sequence alignment was performed using Clustal Omega75, and aligned sequences were coloured based on sequence identity (identity parameter 99%), using the BLOSUM6276 colour scheme.

### 4.8. Polymerase Chain Reaction (PCR) and Gel Electrophoresis

To validate the presence and expression of *appB* and *appC* in the *M. smegmatis* genome, genomic DNA and cDNA from *M. smegmatis* WT and *bd*-KO PCR was subjected to PCR amplification using suitable primers (Appendix A). The Platinum™ Taq DNA Polymerase kit (Invitrogen™ (Waltham, MA, USA)) was utilised for amplification using following temperature parameters: initial denaturation (180 s, 94 °C), was followed by 30 cycles consisting of 30 s denaturation (94 °C), 30 s annealing (57 °C) and 30 s extension (72 °C). To visualise the PCR products amplified PCR product was run on 2.5% agarose gel.

### 4.9. Preparation of Membrane Fraction

Membrane fractions from *M. smegmatis* were prepared based on a previously described procedure [60]. Briefly, *M. smegmatis* WT and *bd*-KO were harvested from stationary phase (~3 OD) and centrifuged at 6000× *g* for 20 min. The pellet was washed with phosphate-buffered saline (PBS, pH 7.4) and centrifuged at 6000× *g* for 20 min. Each 5 g of cells (wet weight) was re-suspended in 10 mL of ice-cold lysis buffer (10 mM HEPES, 5 mM MgCl_2_ and 10% glycerol at pH 7.5) including protease inhibitors (complete, EDTA-free; protease inhibitor cocktail tablets from Roche). Lysozyme (1.2 mg/mL), deoxyribonuclease I (1500 U, Invitrogen), and MgCl2 (12 mM) were added and cells were incubated with shaking for one hour at 37 °C. The lysates were passed three times through high-pressure cell homogeniser (Standsted (Harlow, UK)) at 1.4 kb to break the cells. Unbroken cells were removed by three centrifugation steps (6000× *g* for 20 min at 4 °C). The membranes were pelleted by ultracentrifugation at 310,000× *g* for one hour at 4 °C. The pellet was re-suspended in 50 mM MOPS, 100 mM NaCl, 0.025% dodecyl-maltoside, pH 7.0. The protein concentration was determined using the BCA Protein Assay kit (Pierce, Thermo Fisher (Waltham, MA, USA)), as described by the manufacturer. When needed, kanamycin (50 ug/mL final conc.) and Q203 (10 uM final conc.) were added to the cultures. For untreated control, DMSO was added in the same percentage as Q203.

### 4.10. Heme Spectra Analysis

The heme content of membranes fractions isolated from *M. smegmatis* WT and *bd*-KO was measured by the reduced minus oxidised UV-VIS spectrum. The isolated membrane fractions were diluted to the concentration of 2.6 mg/mL in 50 mM MOPS, 100 mM NaCl, 0.025% dodecyl-maltoside, pH 7.0. The samples were then oxidised with 100 μM potassium ferricyanide and the spectrum was recorded at room temperature (Varian Cary 50 UV-Vis Spectrophotometer). Subsequently, a few grains of solid sodium hydrosulfite were dissolved in the sample to measure the spectrum in the reduced state and the difference spectrum (reduced-oxidised) was calculated.

### 4.11. Oxygen Consumption Assay Using Extracellular Flux Analyzer

To measure the oxygen consumption of *M. smegmatis* WT and *bd*-KO the Extracellular Flux Analyzer (Seahorse Biosciences (North Billerica, USA)) was utilised. The cultures were grown to stationary phase in presence or absence of Q203, centrifuged for 1 min at 13,000 rpm to remove the growth media, and then washed twice with 7H9 + 0.02% tylaxapol and finally resuspended in 7H9 medium + 0.02% tylaxapol media. The washed bacteria are then diluted to ~0.06 OD_600_ and 50 ul of diluted bacteria were seeded to XF cell culture microplate which was pre-coated with a Cell-Tak based on Lamprecht et al., 2016. The microplate was then centrifuged for 20 min at 2000× *g* in an Eppendorf centrifuge to immobilise the cells plate. After centrifugation, fresh 7H9 medium was added to each well, then the microplate was incubated at 37 °C for 30 min. Glucose and required inhibitors were added to the injection ports at 10-fold higher final concentration along with suitable solvent controls. The initial oxygen consumption rate was measured for 3 cycles before the addition of glucose (2 mg/mL). After three additional cycles, the inhibitors were added. Data analysis was carried out using Wave Desktop 2.2. software (Extracellular Flux Analyzer (Seahorse Bioscience (North Billerica, MA, USA))).

### 4.12. Statistical Analysis

All experimental data are represented as mean ± SD that was determined using the two-way ANOVA test followed by Bonferroni post-tests in GraphPad Prism v7.0 software. *p* values < 0.05 were considered statistically significant and represented as * *p* < 0.05; ** *p* < 0.01 and *** *p* < 0.001.

## Figures and Tables

**Figure 1 ijms-23-10331-f001:**
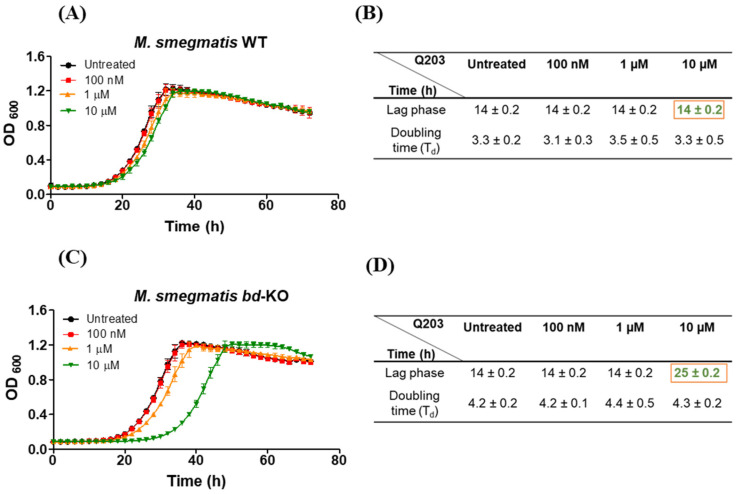
The effect of Q203 on the growth of *M. smegmatis*. Microplate reader-based growth analysis and growth parameters of (**A**,**B**) *M. smegmatis* wild-type (WT) and (**C**,**D**) *M. smegmatis* mutant strain with knocked-out cytochrome *bd* (*bd*-KO) in the presence and absence of indicated Q203 concentrations. The extended lag-phase in presence of Q203 is indicated by a red frame. Mean ± SD of three independent cultures (biological replicates) are plotted.

**Figure 2 ijms-23-10331-f002:**
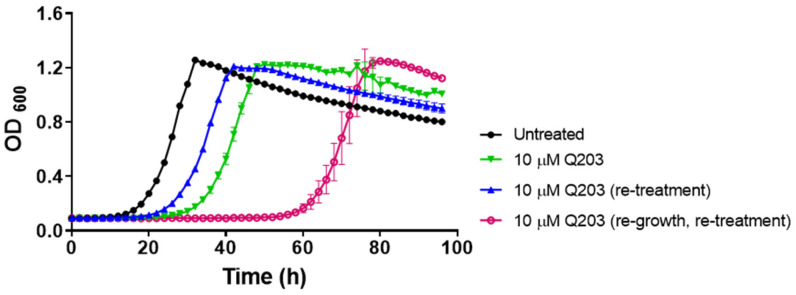
The outgrowth of *M. smegmatis bd*-KO in presence of Q203 is not due to selection of mutant harboring compensatory mutations. After eventual growth resumption in presence of Q203 at ~40 h (green curve), the bacteria were re-treated with Q203 by diluting into fresh 7H9 complete medium containing the drug (blue curve) or re-grown in drug-free 7H9 complete medium for 16 h, and subsequently again diluted into fresh 7H9 complete medium containing the drug. Data are Mean ± SD from two independent experiments (biological replicates), each comprising three technical replicates.

**Figure 3 ijms-23-10331-f003:**
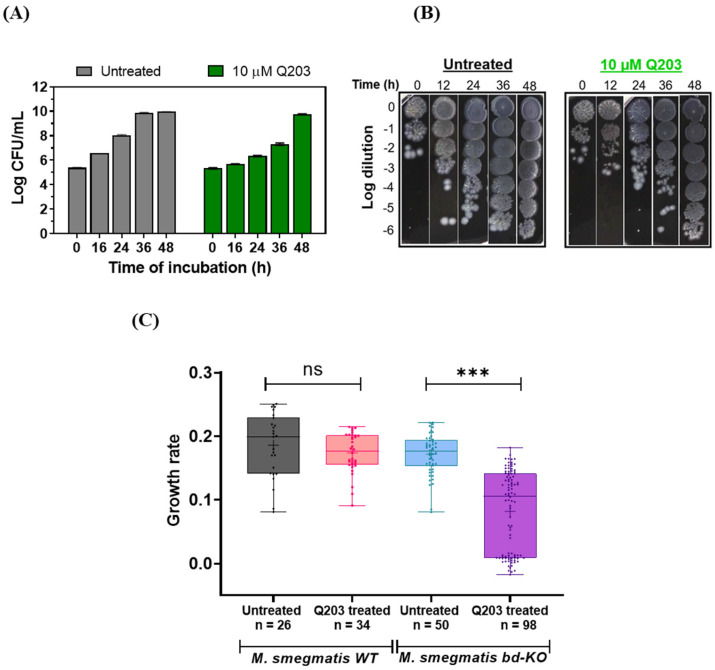
Analysis of variation in adaptability of *Mycobacterium smegmatis* cells to Q203. (**A**) Quantification of surviving *M. smegmatis bd*-KO after exposure to Q203 in 7H9 complete medium for the indicated time. Colony forming units (CFU/mL) were subsequently counted on agar plates after three days of incubation at 37 °C. Data are Mean ± SD from two independent experiments (biological replicates), each comprising three technical replicates. (**B**) Representative dilution series of *M. smegmatis bd*-KO after exposure to Q203 in 7H9 complete medium for the indicated time. Each spot represents a 5 µL aliquot. (**C**) The growth rate of individual *M. smegmatis* WT and *bd*-KO cells was determined using time-lapse microscopy. Cells were imaged for 40 h in 30 min intervals using agarose-pads which were either prepared using 2.5 µM Q203 (treated) or DMSO solvent control (untreated). The determined growth rates of individual cells are indicated by dots in a box-whisker plot. *p* values were calculated by t test in Anova, *** represent *p* < 0.001, ns = non-significant.

**Figure 4 ijms-23-10331-f004:**
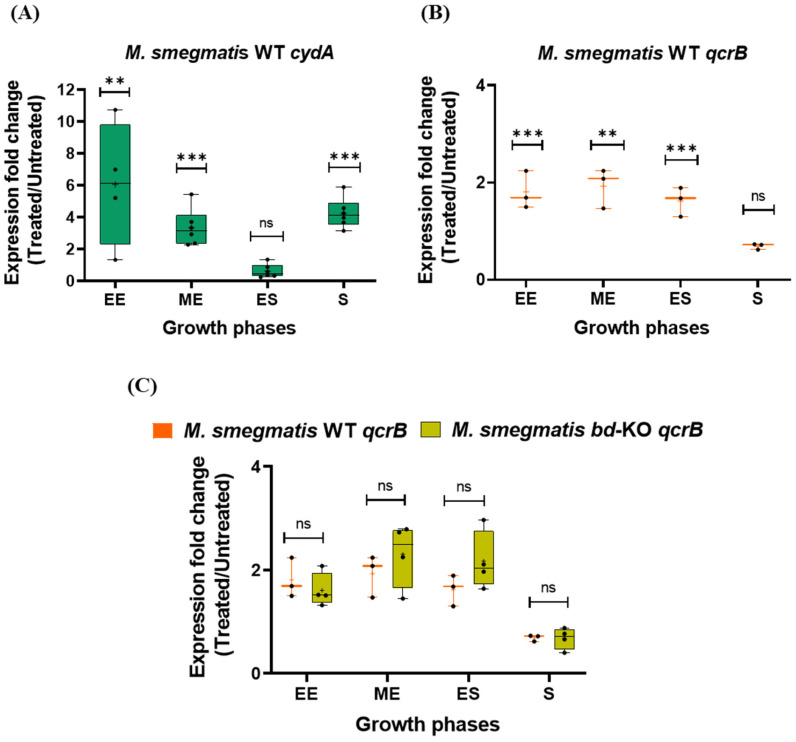
Transcriptional regulation of terminal oxidases of *Mycobacterium smegmatis* in presence of Q203. Fold change in expression of (**A**) the *cydA* gene and (**B**) the *qcrB* gene of *M. smegmatis* WT in Q203-treated versus untreated cultures was assessed in different growth phases by RT-qPCR (using 16S rRNA as control). (**C**) Comparative analysis of fold change expression of *qcrB* gene in *M. smegmatis* WT versus *bd*-KO strain upon treatment with in Q203, as assessed by RT-qPCR. The evaluated growth phases were: early exponential phase (EE); mid-exponential phase (ME); early stationary phase (ES); and stationary phase (S). At least three biological replicates were performed. Each dot represents a biological replicate consisting of two technical replicates. *p* values were calculated with unpaired *t*-test between treated versus untreated culture for panel A and B. For panel C, *p* values was calculated for *M. smegmatis* WT versus *bd*-KO. **, *p*-value ≤ 0.01; *** *p* ≤ 0.001; ns, non-significant.

**Figure 5 ijms-23-10331-f005:**
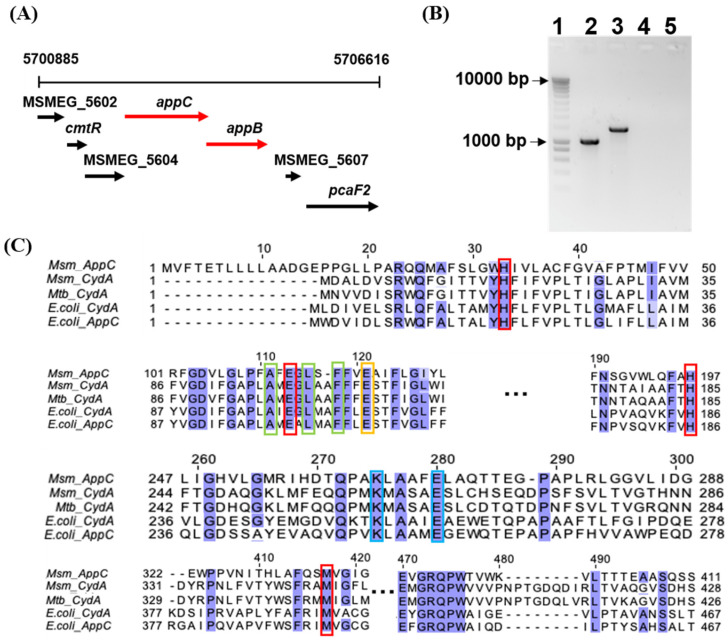
The putative second cytochrome *bd* (AppCB variant) in *Mycobacterium smegmatis*. (**A**) Genomic location of the putative AppCB variant encoding genes *appB* and *appC* in *M. smegmatis*. Putative AppCB variant genes as well as up/downstream genes are depicted as red and black arrows, respectively. (**B**) Validation of *appB* and *appC* expression in *M. smegmatis*. The product is amplified using the cDNA of *appB* and *appC* and appropriate primers. Lane 1, DNA ladder marker; lane 2, amplified *appB*; lane 3, amplified *appC*; lane 4 and 5, negative control (containing *appB* and *appC* primers, but without cDNA). (**C**) Multiple alignment of partial amino acid sequences of *M. smegmatis* (Msm) AppC, Msm CydA, *M. tuberculosis* (Mtb) CydA, *E. coli* AppC, and CydA. Alignment was performed using Clustal Omega, via the online tool Jalview. Protein sequences were derived from the UniProt database. Identical residues were colored using the BLOSUM62 color scheme. Conserved residues involved in heme-binding, putative oxygen channel, proton transfer from cytosol to heme d, or quinol binding are indicated with red, green, orange, and blue boxes, respectively. For alignments of the complete amino acid sequences, please refer to Appendix A.

**Figure 6 ijms-23-10331-f006:**
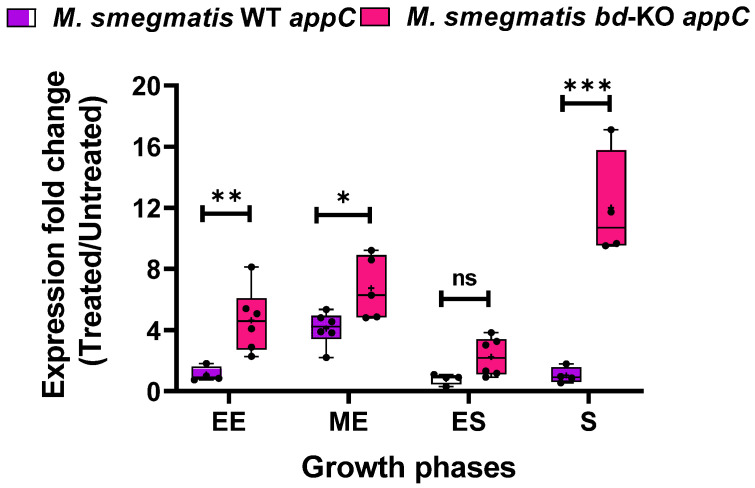
Expression analysis of the *appC* gene of *Mycobacterium smegmatis* WT and *bd*-KO in presence of Q203. Comparative analysis of fold change expression of *appC* gene in *M. smegmatis* WT versus *bd*-KO strain upon treatment with in Q203, as assessed by RT-qPCR (using 16S rRNA as normalising control). The evaluated growth phases were: early exponential phase (EE); mid-exponential phase (ME); early stationary phase (ES); and stationary phase (S). At least three biological replicates were performed. Each dot represents a biological replicate consisting of two technical replicates. *p* values was calculated with unpaired *t*-test for *M. smegmatis* WT versus *bd*-KO. * represents *p*-value ≤ 0.05; **, *p*-value ≤ 0.01; *** *p* ≤ 0.001; ns, non-significant.

**Figure 7 ijms-23-10331-f007:**
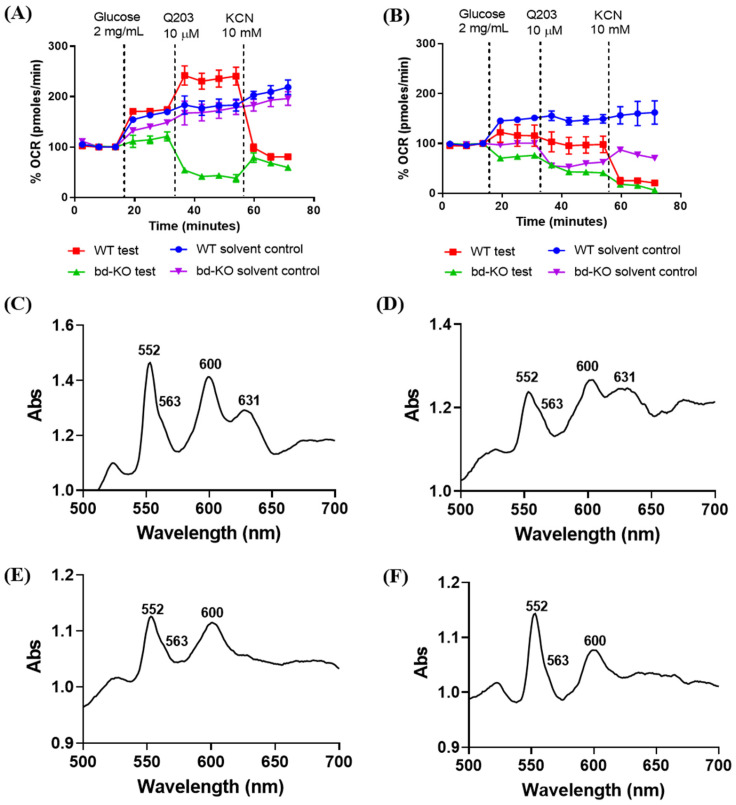
Impact of Q203 on the respiration and heme content of *Mycobacterium smegmatis*. The oxygen consumption activity of *M. smegmatis* WT and *bd*-KO strain grown with (**A**) DMSO (control) or (**B**) 10 µM Q203. The cells were harvested in stationary phase and immobilised onto a microplate of an Extracellular Flux Analyzer. The oxygen consumption rate (OCR) was measured for three cycles, subsequently glucose (2 mg/mL), Q203 (10 µM) and KCN (10 mM) were added at the indicated time points. The OCR after addition of glucose to energise the bacteria (typically 60 pmol/min for WT and 40 pmol/min for *bd*-KO grown in presence of DMSO and 150 pmol/min and 60 pmol/min for *bd*-KO grown in presence of Q203) was set to 100%. Three biological replicates were performed, results from one representative biological replicate are shown, error bars show standard deviations of values from two technical replicates/wells. (**C**–**F**) Reduced-minus-oxidised spectra of membrane fractions isolated from *M. smegmatis* WT and *bd*-KO grown in presence of (**C**,**E**) DMSO (untreated control) and (**D**,**F**) 10 µM Q203, respectively. Samples were reduced by the addition of sodium dithionite and oxidised using potassium ferricyanide.

## Data Availability

All the data presented here are included in the manuscript and Appendix A.

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
