# Peer review of "Response of Mycobacterium smegmatis to the Cytochrome bcc Inhibitor Q203"

_ijms, 2022, doi:10.3390/ijms231810331_

Round 1
Reviewer 1 Report
The authors presented a nice and well-written manuscript about Q203 and its action mechanisms, using M. smegmatis as the research platform since it is naturally not sensitive to the drug.
Their findings are relevant and made an excellent contribution to the understanding of the role of the target. Yet, the authors searched for another potential gene that could be involved in the adaptation to Q203, discarding its role in the process.
For me, the manuscript is ready to publish, as it is.
In the Sup. material, Figure S1 A must have corrected the concentration labels (the text box is hiding the numbers).
Author Response
We thank the reviewer for his/her positive comments. In the revised version of the supplementary file, we have updated the concentration labels in Figure S1 based on the referee's comment.
Reviewer 2 Report
In this manuscript, Chauhan et al. evaluated the response of M. smegmatis to the cytochrome bc1 inhibitor Q203. In my opinion, this manuscript is sounded and reveals novel insights into the adaptation of M. smegmatis to Q203 treatment regarding its terminal oxidases. The authors suggested that M. smegmatis can adapt to Q203 treatment by regulating the expression of its terminal oxidases, including a cytochrome bd oxidase isoform encoded by AppB and AppC. Here are my comments:
1- Lines 83-85: In this study, did the authors use a ∆cydA, ∆cydB or ∆cydAB strain? Reference 28 reported several KO mutants. It will be necessary to review this manuscript to clarify which strain was used as M. smegmatis bd-KO strain.
2- Figuresure 1 and 2: The authors showed that Q203 inhibition in the ∆bd oxidase strain can be outgrown and delay the stationary phase. I will highly suggest that the authors add a complemented ∆bd oxidase strain. These results can be due to a polar effect on other components of the cydAB (e.g. cydDC).
3- Figure 3A: please change the X-axis title to “time of incubation” instead of “time of treatment.”
4- Lines 147-150: Did the authors consider that the variability in the growth rate of the bd-KO strain can be caused by the Q203 concentration used for the time-lapse experiments rather than adaptation to the drug? The authors used 2.5uM, which equals the MIC concentration in the bd-KO strain. Following the MIC definition, not all bacteria will be inhibited by Q203 at this concentration, which would explain the scattered results obtained for the bd-KO strain exposed to Q203. Please provide further explanation.
5- Figure 4: Which criteria were the different growth phases determined? Please provide further explanation in the manuscript.
6- Lines 214-219: The authors identified a putative second cytochrome bd encoded by AppC and AppB. Did the authors identify a second cydDC operon in M. smegmatis?
7- Did the authors try to inactivate AppB or AppC in M. smegmatis?
Author Response
We thank to referee for his/her positive comments on our manuscript. Here we address the referee's remarks point by point:
Point 1: Lines 83-85: In this study, did the authors use a ∆cydA, ∆cydB or ∆cydAB strain? Reference 28 reported several KO mutants. It will be necessary to review this manuscript to clarify which strain was used as M. smegmatis bd-KO strain.
Response 1: Indeed reference 28 describes several Msm bd-KO strains. The M. smegmatis bd-KO strain that we used for our study is the M. smegmatis mc2 155 cydA::aph. In this strain, the cydA gene is inactivated by a kanamycin resistance cassette (Ref 28). We added this information to the Materials and Methods section (lines 418- 420 of the revised manuscript).
Point 2: Figuresure 1 and 2: The authors showed that Q203 inhibition in the ∆bd oxidase strain can be outgrown and delay the stationary phase. I will highly suggest that the authors add a complemented ∆bd oxidase strain. These results can be due to a polar effect on other components of the cydAB (e.g. cydDC).
Response 2: We appreciate the suggestion by the reviewer. Indeed complementation can strengthen the message from knock-out experiments. However, in the case of cytochrome bd, attempts to introduce plasmids expressing cytochrome bd into an M. smegmatis bd-KO strain were either not successfully reported (Reference 26) or led to only a low degree of complementation (References 28). Regarding the cydCD genes which are located downstream of the cydAB genes: it has been shown that in M. smegmatis the cydAB genes and cydDC genes form separate operons. Therefore, we regard a downstream effect of a cydA gene knockout on cydDC as unlikely (Aung et al 2014, J Bacteriol, 196; 3091-3097).
Point 3: Figure 3A: please change the X-axis title to “time of incubation” instead of “time of treatment.”
Response 3: We thank the reviewer for this suggestion and we have edited the axis title accordingly.
Point 4: Lines 147-150: Did the authors consider that the variability in the growth rate of the bd-KO strain can be caused by the Q203 concentration used for the time-lapse experiments rather than adaptation to the drug? The authors used 2.5 uM, which equals the MIC concentration in the bd-KO strain. Following the MIC definition, not all bacteria will be inhibited by Q203 at this concentration, which would explain the scattered results obtained for the bd-KO strain exposed to Q203. Please provide further explanation.
Response 4: We agree with the reviewer, indeed at this chosen concentration the bacterial growth is not fully inhibited. However, what we observed is that the bd-KO strain in the presence of Q203 displays enhanced variability, both compared to the WT strain in the presence of Q203 and compared to the bd-KO strain in absence of Q203. We have added this information to the relevant paragraph of the Results section (lines 149-151).
Point 5: Figure 4: Which criteria were the different growth phases determined? Please provide further explanation in the manuscript.
Response 5: We thank the reviewer for pointing out the insufficient description in our initial submission. We have now added the information on how growth phases were determined to the Materials and Methods section (lines 434-443 of the revised manuscript).
Point 6: Lines 214-219: The authors identified a putative second cytochrome bd encoded by AppC and AppB. Did the authors identify a second cydDC operon in M. smegmatis?
Response 6: We did not identify a second cydDC operon in the M. smegmatis genome. Downstream of the appC and appB genes are genes that bear no resemblance to cydDC genes. If AppB and AppC represent a bonafide second cytochrome bd in M. smegmatis then it seems to be dependent on the characterized CydDC. This situation would be similar to E. coli, where two cytochrome bd variants are known but only one cydDC.
Point 7: Did the authors try to inactivate AppB or AppC in M. smegmatis?
Response 7: This is an exciting suggestion but we think it will exceed the scope of our current manuscript. We regard such an inactivation experiment as exquisitely suitable for a follow-up study.